# The Optimal Cultivar × Sowing Date × Plant Density for Grain Yield and Resource Use Efficiency of Summer Maize in the Northern Huang–Huai–Hai Plain of China

Lichao Zhai [1], Lihua Zhang [1,*], Haipo Yao [1], Mengjing Zheng [1], Bo Ming [2], Ruizhi Xie [2], Jingting Zhang [1], Xiuling Jia [1] and Junjie Ji [1]

[1] Institute of Cereal and Oil Crops, Hebei Academy of Agricultural and Forestry Science/Key Laboratory of Crop Cultivation Physiology and Green Production of Hebei Province/Scientific observing and Experimental Station of Crop Cultivation in the North China, Ministry of Agriculture and Rural Affairs, Shijiazhuang 050035, China; zhailichao@163.com (L.Z.); yhpkuaile@126.com (H.Y.); zhengmj96@126.com (M.Z.); jingting58@126.com (J.Z.); jiaxiuling2013@163.com (X.J.); 13503211982@163.com (J.J.)
[2] Institute of Crop Sciences, Chinese Academy of Agricultural Sciences/Key Laboratory of Crop Physiology and Ecology, Ministry of Agriculture and Rural Affairs, Beijing 100081, China; obgnim@163.com (B.M.); xieruizhi@caas.cn (R.X.)
* Correspondence: lnzlh@126.com; Tel.: +86-311-87-670-620

**Abstract:** In order to explore the optimal cultivar × sowing date × plant density for summer maize (*Zea mays* L.) in the Northern Huang–Huai–Hai (HHH) Plain of China, field experiments were conducted over two consecutive years (2018–2019) on a loam soil in the Northern HHH Plain. A split–split plot design was employed in this study, and the main plots included three cultivars (HM1: early-maturing cultivar; ZD958: medium-maturing cultivar; DH605: late-maturing cultivar); subplots consisted of three sowing dates (SD1: June 10; SD2: June 17; SD3: June 24); sub-sub plots include two plant densities (PD1: $6.75 \times 10^4$ plants ha$^{-1}$; PD2: $8.25 \times 10^4$ plants ha$^{-1}$). The results showed that the effects of cultivar and plant density on grain yield of summer maize were not significant, and the sowing date was the major factor affecting the grain yield. Delayed sowing significantly decreased the grain yield of summer maize, this was due mainly to the reduced kernel weight, which is associated with the lower post-anthesis dry matter accumulation. Moreover, radiation use efficiency (RUE), temperature use efficiency (TUE), and water use efficiency (WUE) were significantly affected by cultivar, sowing date, and plant density. Selecting early- and medium-maturing cultivars was beneficial to the improvements in RUE and TUE, and plants grown at earlier sowing with higher plant density increased the RUE and TUE. The interactive analysis of cultivar × sowing date × plant density showed that the optimum grain yields of all tested cultivars were observed at SD1-PD2, and the optimum RUE and TUE for HM1, ZD958, and DH605 were observed at SD1-PD2, SD2-PD2, and SD2-PD2, respectively. The differences in the optimum grain yield, RUE, and TUE among the tested cultivars were not significant. These results suggested that plants grown at earlier sowing with reasonable dense planting had benefits of grain yield and resource use efficiency. In order to adapt to mechanized grain harvesting, early-maturing cultivar with lower grain moisture at harvest would be the better choice. Therefore, adopting early-maturing cultivars grown with earlier sowing with reasonably higher plant density would be the optimal planting pattern for summer maize production in the Northern HHH Plain of China in future.

**Keywords:** summer maize; cultivar; sowing date; plant density; grain yield; resource use efficiency



## 1. Introduction

A growing body of research indicates that climate change has adverse effects on crop production [1,2], which poses a great challenge to food security worldwide [3,4]. The Huang–Huai–Hai (HHH) Plain is one of the most important agricultural regions for maize

production in China, contributing about 33% of the maize produced by the entire nation [5]. In recent years, the frequent heat stress resulted by warming climate has been the most important contributor in reduced maize grain yields [6,7], especially in the Northern HHH Plain. Therefore, optimizing culture and management practices to adapt the local summer maize production is urgently needed.

Selecting adapted maize cultivars is an effective way to cope with some of the adverse effects of climate change [8–10]. Some previous studies suggested that adapted late-maturing maize cultivars could effectively offset the negative impacts of a warming climate on crop productivity [10]. However, the winter wheat–summer maize double-cropping system is the main cropping technique in the Northern HHH Plain, the growth period of summer maize was relatively narrow, and adopting late-maturing maize cultivar needs to postpone the harvesting time until at least mid–late October, which is not conducive to the timely sowing of winter wheat. Therefore, further studies are needed to explore the suitability of planting later-maturing maize cultivars in the Northern HHH Plain.

The sowing date is a key aspect of crop management which is frequently manipulated to adjust the timing and occurrence of crop phenological phases according to the environmental conditions [11–13]. Previous studies have reported that adjustments of the sowing date could increase grain yield and water use efficiency of maize in a rain-fed farming system in arid and semiarid areas [14], and late-sown maize with adaptive culture practices could improve maize grain yields [15]. However, any delay in sowing time diminished the degree of synchronization between peak solar radiation and maximum green leaf area index for maize hybrid varieties [16]. Reductions in grain yield due to early or late sowing have been well documented in the literature during the past 10 years [15,17,18]. Thus, the optimal sowing date for maize depends on both the specific region and cultivars.

With enhancements in the density tolerance of modern maize varieties, increasing plant density reasonably is one of the most important agronomic practices for increasing the grain yield potential and resource use efficiency of maize worldwide [19,20]. In the HHH Plain, the average plant density adopted by smallholder farmers is about 62,000 pl ha$^{-1}$ [21], which is much lower than the average plant density of the maize belt in the United States [22,23]. Previous studies have demonstrated that reasonably increasing the plant density can increase the potential capacity of the crop canopy to capture resources, including solar radiation, water, and nutrients [24–26]. However, under unreasonably close planting conditions, leaf shading can lead to poor canopy ventilation and light penetration, resulting in thin stems, increased maize lodging, and decreased dry matter production, which ultimately lead to lower grain yield [27,28]. Therefore, the planting density should be adjusted depending on the density tolerance of cultivars and the climatic conditions of a region.

To improve grain yield and minimize the adverse effects of climate change, maize producers in the Northern HHH Plain have already adjusted the sowing date, alternated maize cultivars, or used these two measures in combination. Moreover, in order to improve the production efficiency of summer maize, mechanized grain harvesting for maize become popular in the HHH Plain; however, studies on improving grain yield and resource use efficiency by optimizing the cultivar, sowing date, and plant density of summer maize under such backgrounds are limited. In this context, we hypothesize that the grain yield and resource use efficiency of summer maize could be improved by optimizing cultivars, sowing date, and plant density, and this optimized culture practice also meets the needs of mechanized grain harvesting. In order to verify this hypothesis, field experiments were conducted with objectives to (1) investigate the individual and combined effects of cultivar, sowing date, and plant density on grain yield and resource use efficiency of summer maize, and (2) determine the optimal cultivar × sowing date × plant density for grain yield and resource use efficiency of summer maize in the Northern HHH Plain.

## 2. Materials and Methods

### 2.1. Experimental Site

Field experiments were conducted during the 2018 and 2019 growing seasons at Guantao Experimental Station (36°72′ N, 115°37′ E) of the Hebei Academy of Agriculture and Forest Science, which is located in Handan, Hebei Province, on the Northern HHH Plain (Figure 1). Guantao is in a warm temperate zone with a semi-humid continental monsoon climate, The mean annual temperature and precipitation amounts of the experimental site are 12.4 °C and 600 mm, respectively. Meteorological data from the experimental sites were obtained from the China Meteorological Data Sharing Service System (http://www.cdc.nmic.cn, accessed on 5 May 2020), and the monthly meteorological data during the two growing seasons are shown in Figure 2, with precipitation during the 2018 and 2019 growing seasons being 234.8 and 262.4 mm, respectively. Over the past decade in this region, the main cropping system has been winter wheat–summer maize rotation. The primary soil texture is loam, the basic soil fertility in the upper 0–20 cm of the soil profile before sowing is detailed in Table 1; the soil bulk density and porosity of the 0–20 cm soil layer were 1.53 g m$^{-3}$ and 42.3%, respectively.

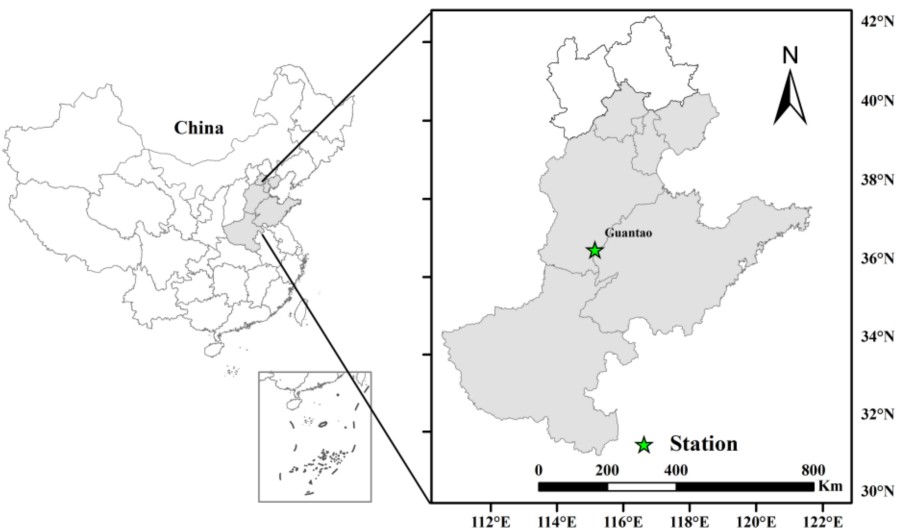

**Figure 1.** Location of the Huang–Huai–Hai (HHH) Plain in China (**left**) and of the experimental site within the HHH Plain (**right**).

### 2.2. Experimental Design and Field Management

The early-maturing maize cultivar Huamei 1 (HM1), the medium-maturing maize cultivar Zhengdan 958 (ZD958), and late-maturing cultivar Denghai 605 (DH605) were used as materials; these cultivars have been planted widely across the HHH Plain in recent years. HM1 was characterized with a semi-compact plant type, and ZD958 and DH605 were characterized with a compact plant type. Field experiments were conducted using a split–split plot design during the two growing seasons. Maize cultivars (HM1, ZD958, and DH605) were the main plot factor; the sowing dates (SD1: June 10, SD2: June 17, and SD3: June 24) were the sub-plot factors; and plant density (PD1: $6.75 \times 10^4$ plants ha$^{-1}$ and PD2: $8.25 \times 10^4$ plants ha$^{-1}$) was the sub-sub plot factor. The chemical fertilizers N, $P_2O_5$, and $K_2O$ were applied at sowing in amounts of 270, 144, and 144 kg ha$^{-1}$, respectively. The chemical fertilizers included 720 kg ha$^{-1}$ of compound fertilizer (N:$P_2O_5$:$K_2O$, 15%:15%:15%), 368 kg ha$^{-1}$ of controlled release urea (44% N), 100 kg ha$^{-1}$ of calcium phosphate (40% $P_2O_5$), and 100 kg ha$^{-1}$ of potassium sulfate (60% $K_2O$). Seeds were planted manually with a row spacing of 60 cm; the experimental plot had dimensions of 10 m × 10 m, each treatment included three replications, and there was a 1.0 m isolation area between each plot. The irrigation amount was applied according to the local precip-

itation. Weeds, insects, and diseases were controlled in a timely manner based on local agronomic practices to eliminate their negative effects on maize growth and grain yield.

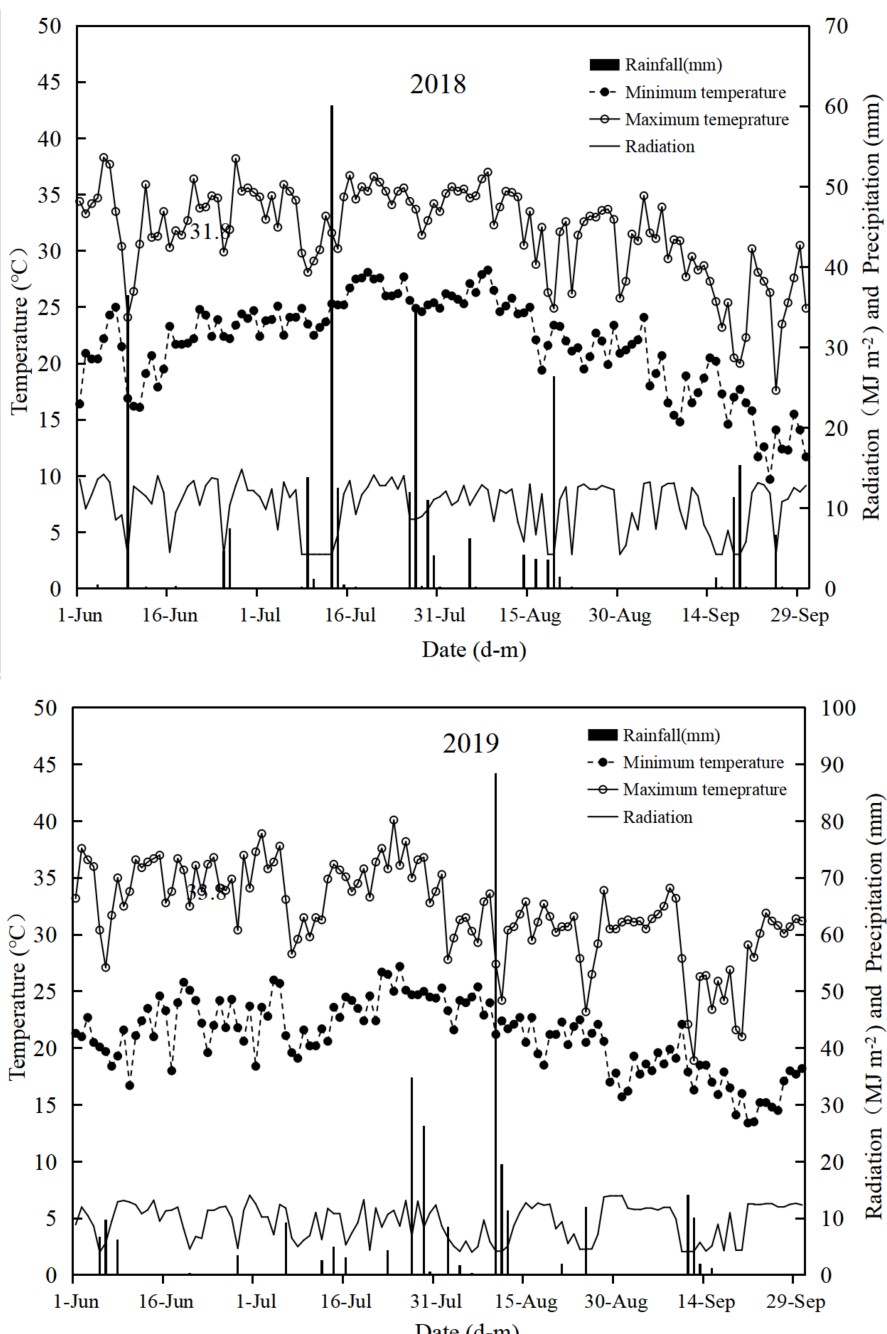

**Figure 2.** Daily solar radiation, rainfall, and maximum and minimum air temperature at the experimental station.

**Table 1.** The basic soil characteristics of the experimental sites before sowing.

| Year | Organic Matter (%) | Total Nitrogen (%) | Available Nitrogen (mg kg$^{-1}$) | Available Phosphorus (mg kg$^{-1}$) | Available Potassium (mg kg$^{-1}$) |
|---|---|---|---|---|---|
| 2018 | 1.24 | 0.12 | 100.66 | 41.55 | 135.15 |
| 2019 | 1.39 | 0.13 | 144.74 | 43.66 | 195.19 |

### 2.3. Sampling and Measurements

Dry matter accumulation (DMA). To measure DMA, three successive uniform plants were selected manually in the middle of each plot and cut at ground level at both anthesis and physiological maturity. The plants were separated into stalks, leaves, sheaths, tassels, and ears, and oven-dried. Post-anthesis DMA was estimated as the difference in biomass between the physiological maturity and anthesis results.

Radiation use efficiency (*RUE*). *RUE* was calculated using the following equation:

$$RUE = \frac{GY}{Q}$$

where *GY* is the grain yield (kg ha$^{-1}$) and *Q* is the accumulated solar radiation (MJ m$^{-2}$) during the crop growth period. The solar radiation, *Q*, was calculated according to the following equation [29]:

$$Q = Q_0 \times \left( a + \frac{bS}{S_0} \right)$$

where *Q* is the total accumulated solar radiation (MJ m$^{-2}$), $Q_0$ is the astronomical radiation (MJ m$^{-2}$), S is the actual sunshine hours (h), $S_0$ is the possible sunshine hours (h), and *a* and *b* are correlation coefficients, which were 0.248 and 0.752, respectively.

Temperature use efficiency (*TUE*). *TEU* reflects the cumulative temperature production efficiency, calculated using the following equation:

$$TUE = \frac{GY}{GDD}$$

where *GY* is the grain yield (kg ha$^{-1}$) and *GDD* is the effective accumulated temperature during the crop growth period. *GDD* was calculated as follows:

$$GDD = \sum \frac{T_{\max} + T_{\min}}{2} - T_{base}$$

where $T_{max}$ and $T_{min}$ are the daily maximum and minimum temperatures, respectively, and $T_{base}$ is the base temperature for maize (10 °C) [30].

Water use efficiency (*WUE*). *WUE* was calculated using the following equation:

$$WUE = \frac{GY}{ET_a}$$

where *GY* is the grain yield (kg ha$^{-1}$) and $ET_a$ is water consumption over the entire growing season. $ET_a$ was calculated using the following water balance equation [31]:

$$ET_a = P + I + \Delta SWD - R - D + CR$$

where *P* (mm) is precipitation, *I* (mm) is irrigation, $\Delta SWD$ (mm) is soil water extraction based on the difference between sowing and maturity; soil water contents were measured with an oven-drying method, and soil samples were collected using a soil auger with three replicates (4.5 cm diameter at 20 cm increments to a depth of 180 cm). *R* is surface runoff, *D* is drainage below the 200 cm soil profile, and *CR* is capillary rise into the root zone, which was negligible because the groundwater table was more than 37 m deep at the experimental site. *R* and *D* are also considered negligible on the North China Plain [32].

Grain yield and yield components. At harvest, grain yield (GY) was determined following grain black layer formation by hand, harvesting all ears from a 5 m × 2 m site in the middle rows of each plot under the condition of 14% grain moisture content, and the ear number (EN) per unit area of each treatment was calculated from the harvested ears. From the harvested ears, 15 were chosen to measure ear characteristics after 20 days of natural air drying. The ear characteristics included row number, kernels per row, and kernels per ear

(KPE). After attaining these measurements, kernels were threshed by a grain thresher, and the 300-kernel weight (KW) was measured and converted to the 1000-kernel weight (KW).

### 2.4. Data Analysis

SPSS software (ver. 19.0; SPSS Inc., Chicago, IL, USA) was used to perform analysis of variance (ANOVA; univariate general linear model); cultivar, sowing date, and plant density were the fixed factors, and a random block design was used. Graphs were plotted in either SigmaPlot (ver. 12.0; Systat Software, San Jose, CA, USA) or Excel 2016 (Microsoft Corp., Redmond, WA, USA). All parameters were tested for normality and found to be normally distributed using the Shapiro–Wilk W test [33]. Homogeneity of variance was assessed using Levene's test, appropriate transformations were applied to response variables that violated assumptions, and back-transformed data were reported. Comparisons of treatment means were performed using the post hoc Tukey's HSD test at the 0.05 level of probability.

## 3. Results

### 3.1. Grain Yield and Yield Components

Analysis of variance showed that only the effect of sowing date on GY was significant (Tables 2 and 3). Delaying the sowing date significantly reduced the GY, and increasing the plant density improved the GY. The interactive analysis of cultivar, sowing date, and plant density showed that the maximum GYs for all tested maize cultivars were obtained at SD1-PD2, with the exception of ZD958 in 2018, and the maximum GYs of HM1, ZD958, and DH605 was not significantly different from each other. In addition, ANOVA revealed that the effects of cultivar, sowing date, and plant density on yield components were significant in most cases (Tables 2 and 4). The KNP of HM1 was significantly higher than that of ZD958 and DH605, but the kernel weight of HM1 and ZD958 was lower than that of DH605. Delaying the sowing date and increasing the plant density reduced both KNP and KW. The interactive analysis of cultivar, sowing date, and plant density showed that only significant differences were observed for KW.

**Table 2.** Analysis of variance on grain yield, yield components, and dry matter accumulation.

| Source of Variation | *p* Value for | | | | | | |
|---|---|---|---|---|---|---|---|
| | Ear Number | Kernels per Ear | Kernel Weight | Grain Yield | Pre-Anthesis DM | Post-Anthesis DM | Total DM |
| Year (Y) | ns | <0.0001 | 0.010 | <0.0001 | <0.0001 | 0.001 | ns |
| Cultivar (C) | ns | <0.0001 | <0.0001 | ns | <0.0001 | <0.0001 | <0.0001 |
| Sowing date (SD) | 0.002 | <0.0001 | <0.0001 | <0.0001 | <0.0001 | <0.0001 | <0.0001 |
| Plant density (PD) | <0.0001 | <0.0001 | <0.0001 | 0.050 | <0.0001 | 0.045 | 0.001 |
| Y × C | ns | ns | 0.006 | ns | ns | ns | 0.011 |
| Y × SD | 0.014 | <0.0001 | <0.0001 | <0.0001 | 0.002 | <0.0001 | <0.0001 |
| Y × PD | ns | ns | ns | ns | ns | ns | ns |
| C × SD | ns | 0.017 | <0.0001 | <0.0001 | 0.011 | <0.0001 | <0.0001 |
| C × PD | ns | ns | ns | 0.009 | ns | ns | ns |
| SD × PD | ns | ns | ns | ns | ns | ns | ns |
| Y × C × SD | ns | 0.002 | 0.011 | <0.0001 | ns | ns | 0.016 |
| Y × C × PD | ns | ns | <0.0001 | ns | 0.005 | ns | ns |
| Y × SD × PD | ns | ns | ns | ns | ns | 0.009 | 0.017 |
| C × SD × PD | ns | ns | <0.0001 | ns | 0.037 | 0.008 | <0.0001 |
| Y × C × SD × PD | ns | ns | <0.0001 | ns | 0.034 | ns | ns |

Note: ns indicate no significant difference was observed.

### 3.2. Dry Matter Accumulation (DMA)

The ANOVA showed significant effects of cultivar, sowing date, and plant density on pre-anthesis, post-anthesis, and total DMA (Tables 2 and 4). For pre-anthesis DMA, the tested cultivars was significantly different from each other, with the order ZD958 > DH605 > HM1; the early seeding date was beneficial to pre-anthesis DMA. For post-anthesis and total DMA, ZD958 and DH605 were significantly higher than that of HM1, and early sowing also increased the post-anthesis and total DMA. In both growing seasons, the interactive analysis showed that the maximum DMA of HM1 and ZD958 were observed at SD1-PD2, DH605 reached its maximum at SD1-PD1, and the maximum DMA of DH605 was significantly higher than that of HM1 and ZD958.

**Table 3.** Effects of cultivar, sowing date and plant density on grain yield and yield components of maize cultivars differing in maturity.

| Year | Cultivar | Sowing Date | Plant Density | Grain Yield (t ha$^{-1}$) | EN | KNP | 1000-KW (g) |
|---|---|---|---|---|---|---|---|
| 2018 | HM1 | SD1 | PD1 | 9.69 ± 0.21bcde | 6.67 ± 0.08b | 555.2 ± 9.9a | 299.3 ± 8.7j |
| | | | PD2 | 10.32 ± 0.36abc | 8.06 ± 0.08a | 489.0 ± 1.7bc | 302.0 ± 5.1ij |
| | | SD2 | PD1 | 9.34 ± 0.12def | 6.50 ± 0.19b | 508.1 ± 14.5b | 392.4 ± 8.2a |
| | | | PD2 | 9.62 ± 0.13cde | 7.89 ± 0.05a | 442.5 ± 23.4cdefg | 311.4 ± 6.2ghij |
| | | SD3 | PD1 | 7.36 ± 0.24i | 6.60 ± 0.17b | 428.9 ± 13.5defgh | 327.2 ± 11.2fgh |
| | | | PD2 | 8.59 ± 0.17fg | 7.95 ± 0.07a | 401.9 ± 7.8fgh | 325.0 ± 2.4fghi |
| | ZD958 | SD1 | PD1 | 9.55 ± 0.23cde | 6.80 ± 0.28b | 473.7 ± 12.6bcd | 333.9 ± 1.6efg |
| | | | PD2 | 9.53 ± 0.29cde | 8.06 ± 0.08a | 421.5 ± 16.7efgh | 319.5 ± 6.7fghij |
| | | SD2 | PD1 | 10.27 ± 0.34abc | 6.68 ± 0.02b | 440.1 ± 24.6defg | 365.2 ± 4.1bcd |
| | | | PD2 | 9.61 ± 0.03cde | 8.02 ± 0.08a | 388.9 ± 18.8ghi | 355.7 ± 6.4cde |
| | | SD3 | PD1 | 9.51 ± 0.30cde | 6.65 ± 0.05b | 414.5 ± 17.4fgh | 342.8 ± 4.6def |
| | | | PD2 | 9.13 ± 0.26ef | 8.04 ± 0.01a | 355.4 ± 20.6i | 327.5 ± 5.8fgh |
| | DH605 | SD1 | PD1 | 10.56 ± 0.43ab | 6.77 ± 0.12b | 467.9 ± 4.8bcde | 383.3 ± 4.5ab |
| | | | PD2 | 10.65 ± 0.39a | 7.87 ± 0.12a | 421.9 ± 6.4efgh | 375.9 ± 6.1abc |
| | | SD2 | PD1 | 10.15 ± 0.10hi | 6.67 ± 0.00b | 456.5 ± 13.5cdef | 315.2 ± 11.9ghij |
| | | | PD2 | 10.33 ± 0.40abc | 7.98 ± 0.05a | 385.7 ± 13.2hi | 373.3 ± 16.2abc |
| | | SD3 | PD1 | 8.22 ± 0.31gh | 6.81 ± 0.08b | 442.3 ± 6.0cdefg | 326.2 ± 4.7fghi |
| | | | PD2 | 7.64 ± 0.14hi | 7.90 ± 0.05a | 385.8 ± 16.8hi | 307.7 ± 4.1hij |
| 2019 | HM1 | SD1 | PD1 | 11.98 ± 0.12cde | 6.86 ± 0.12cd | 561.0 ± 11.0ab | 333.7 ± 3.8e |
| | | | PD2 | 12.92 ± 0.27abcd | 8.27 ± 0.14a | 494.1 ± 10.3def | 342.2 ± 2.9de |
| | | SD2 | PD1 | 11.82 ± 0.21cde | 6.70 ± 0.03cdef | 531.4 ± 12.5bcd | 342.8 ± 7.1de |
| | | | PD2 | 11.98 ± 0.19cde | 8.09 ± 0.03ab | 470.9 ± 11.5efgh | 340.0 ± 2.8de |
| | | SD3 | PD1 | 10.27 ± 0.59fg | 6.42 ± 0.08f | 579.6 ± 28.3a | 280.9 ± 2.9fg |
| | | | PD2 | 11.28 ± 0.31ef | 8.09 ± 0.17ab | 514.6 ± 14.0cde | 285.5 ± 2.7fg |
| | ZD958 | SD1 | PD1 | 13.04 ± 0.29abc | 6.72 ± 0.11cdef | 481.0 ± 24.5efg | 376.1 ± 6.9b |
| | | | PD2 | 13.24 ± 0.72ab | 8.28 ± 0.03a | 433.8 ± 20.7h | 363.3 ± 11.9bc |
| | | SD2 | PD1 | 12.69 ± 0.29abcd | 6.88 ± 0.03c | 502.0 ± 4.3def | 333.1 ± 5.0e |
| | | | PD2 | 12.63 ± 0.21abcd | 8.08 ± 0.09ab | 462.6 ± 18.1fgh | 322.4 ± 7.6e |
| | | SD3 | PD1 | 9.56 ± 0.09g | 6.56 ± 0.17def | 502.3 ± 3.8def | 285.4 ± 7.6fg |
| | | | PD2 | 10.20 ± 0.24fg | 8.01 ± 0.08b | 431.3 ± 14.9h | 268.4 ± 1.2g |
| | DH605 | SD1 | PD1 | 13.01 ± 0.42abc | 6.73 ± 0.03cde | 506.0 ± 4.0def | 417.0 ± 9.6a |
| | | | PD2 | 13.48 ± 0.59a | 8.06 ± 0.06ab | 447.7 ± 10.9gh | 406.6 ± 3.5a |
| | | SD2 | PD1 | 11.73 ± 0.05de | 6.60 ± 0.03cdef | 555.1 ± 0.5abc | 356.9 ± 6.4cd |
| | | | PD2 | 12.04 ± 0.34bcde | 8.14 ± 0.06a | 479.9 ± 3.2efg | 339.1 ± 7.5de |
| | | SD3 | PD1 | 9.66 ± 0.19g | 6.50 ± 0.03ef | 509.1 ± 3.5de | 299.6 ± 1.0f |
| | | | PD2 | 9.15 ± 0.66g | 7.80 ± 0.11b | 486.7 ± 2.8defg | 279.7 ± 10.9fg |

Note: PD1 and PD2 represent the plant densities of $6.75 \times 10^4$ plants ha$^{-1}$ and $8.25 \times 10^4$ plants ha$^{-1}$, respectively. SD1, SD2, and SD3 represent the sowing dates of June 10, June 17, and June 24, respectively. Different letters in the same column within one year indicate significant differences between treatments ($p < 0.05$).

**Table 4.** Effects of cultivar, sowing date and plant density on pre-anthesis, post-anthesis, and total DMA of summer maize.

| Year | Cultivar | Sowing Date | Plant Density | Pre-Anthesis DMA(t ha$^{-1}$) | Post-Anthesis DMA(t ha$^{-1}$) | Total DMA(t ha$^{-1}$) |
|---|---|---|---|---|---|---|
| 2018 | HM1 | SD1 | PD1 | 5.94 ± 0.08fg | 12.44 ± 0.81bcde | 18.38 ± 0.81fg |
| | | | PD2 | 7.97 ± 0.24bcd | 12.18 ± 0.82bcde | 20.16 ± 0.82def |
| | | SD2 | PD1 | 6.50 ± 0.26ef | 8.96 ± 0.33h | 15.46 ± 0.33i |
| | | | PD2 | 7.76 ± 0.74cd | 9.69 ± 0.26gh | 17.45 ± 0.26gh |
| | | SD3 | PD1 | 5.22 ± 0.09g | 10.79 ± 0.08defgh | 16.01 ± 0.08gh |
| | | | PD2 | 6.18 ± 0.22fg | 11.35 ± 0.40cdefg | 17.53 ± 0.49gh |
| | ZD958 | SD1 | PD1 | 7.54 ± 0.16d | 12.12 ± 0.39bcde | 19.66 ± 0.39def |
| | | | PD2 | 9.31 ± 0.17a | 11.15 ± 1.21cdefg | 20.46 ± 1.21de |
| | | SD2 | PD1 | 8.26 ± 0.44abcd | 10.44 ± 0.33efgh | 18.70 ± 0.33efg |
| | | | PD2 | 8.07 ± 0.06bcd | 11.58 ± 0.94cdefg | 19.65 ± 0.94def |
| | | SD3 | PD1 | 7.65 ± 0.47cd | 12.60 ± 1.01bcd | 20.26 ± 1.01def |
| | | | PD2 | 8.64 ± 0.11abc | 11.55 ± 0.24cdefg | 20.19 ± 0.24def |
| | DH605 | SD1 | PD1 | 8.45 ± 0.13abcd | 18.25 ± 0.52a | 26.69 ± 0.52a |
| | | | PD2 | 8.65 ± 0.32abc | 13.96 ± 0.56b | 22.61 ± 0.56bc |
| | | SD2 | PD1 | 7.38 ± 0.20de | 10.09 ± 0.21fgh | 17.47 ± 0.21gh |
| | | | PD2 | 8.44 ± 0.21abcd | 11.75 ± 0.38cdef | 20.16 ± 0.38def |
| | | SD3 | PD1 | 7.85 ± 0.29cd | 13.16 ± 0.56bc | 21.00 ± 0.56cd |
| | | | PD2 | 9.04 ± 0.18ab | 14.04 ± 0.31b | 23.08 ± 0.31b |

**Table 4.** *Cont.*

| Year | Cultivar | Sowing Date | Plant Density | Pre-Anthesis DM A(t ha$^{-1}$) | Post-Anthesis DM A(t ha$^{-1}$) | Total DM A(t ha$^{-1}$) |
|---|---|---|---|---|---|---|
| 2019 | HM1 | SD1 | PD1 | 5.70 ± 0.20fg | 12.92 ± 0.12defg | 18.63 ± 0.12def |
| | | | PD2 | 6.35 ± 0.12ef | 15.78 ± 0.46ab | 22.13 ± 0.46b |
| | | SD2 | PD1 | 5.60 ± 0.12g | 10.49 ± 0.24gh | 16.09 ± 0.24fg |
| | | | PD2 | 5.03 ± 0.19g | 9.83 ± 0.52h | 14.86 ± 0.52g |
| | | SD3 | PD1 | 4.98 ± 0.10g | 11.07 ± 0.29fgh | 16.05 ± 0.29fg |
| | | | PD2 | 7.23 ± 0.39d | 10.84 ± 0.64fgh | 18.06 ± 0.64ef |
| | ZD958 | SD1 | PD1 | 7.32 ± 0.16cd | 13.85 ± 0.49bcde | 21.17 ± 0.49bcd |
| | | | PD2 | 9.16 ± 0.42a | 15.68 ± 0.34abc | 24.84 ± 0.34a |
| | | SD2 | PD1 | 6.85 ± 0.10de | 15.07 ± 0.47bcd | 21.92 ± 0.47b |
| | | | PD2 | 8.11 ± 0.23dc | 13.44 ± 0.61bcdef | 21.55 ± 0.61bc |
| | | SD3 | PD1 | 7.04 ± 0.26de | 10.82 ± 1.09fgh | 17.87 ± 1.09ef |
| | | | PD2 | 8.82 ± 0.21ab | 10.47 ± 0.62gh | 19.29 ± 0.62cde |
| | DH605 | SD1 | PD1 | 6.91 ± 0.22de | 17.73 ± 0.81a | 24.64 ± 0.81a |
| | | | PD2 | 8.45 ± 0.41ab | 15.21 ± 1.83abcd | 23.66 ± 1.83ab |
| | | SD2 | PD1 | 5.56 ± 0.56fg | 13.00 ± 0.82cdefg | 18.56 ± 0.82def |
| | | | PD2 | 7.57 ± 0.15cd | 13.53 ± 1.08bcdef | 21.10 ± 1.08cd |
| | | SD3 | PD1 | 6.86 ± 0.19de | 12.07 ± 0.99efgh | 18.93 ± 0.99cde |
| | | | PD2 | 8.48 ± 0.33ab | 10.46 ± 1.43gh | 18.95 ± 1.43cde |

Note: PD1 and PD2 represent the plant densities of $6.75 \times 10^4$ plants ha$^{-1}$ and $8.25 \times 10^4$ plants ha$^{-1}$, respectively. SD1, SD2, and SD3 represent the sowing dates of June 10, June 17, and June 24, respectively. Different letters in the same column within one year indicate significant differences between treatments ($p < 0.05$).

### 3.3. Radiation Use Efficiency

Table 5 shows that *RUE* was significantly affected by cultivar, sowing date, and plant density, but their interactive effect on RUE was not obvious (Figure 3 and Table 5). In both growing seasons, early sowing date and higher plant density increased the *RUE* of early-maturing cultivar HM1. However, for medium-maturing cultivar ZD958 and late-maturing cultivar DH605, plants grown at SD2 benefited from improved RUE, and increasing the plant density had no significant effect on *RUE*. The interactive analysis showed that the optimal *RUE*s for HM1, ZD958, and DH601 were observed at SD1-PD2, SD2-PD1, and SD2-PD2, respectively, and these were not significantly different from each other.

**Table 5.** Analysis of variance on *RUE*, *TUE*, and *WUE*.

| Source of Variation | p Value for | | |
|---|---|---|---|
| | *RUE* (kg MJ$^{-1}$) | *TUE* (kg °C$^{-1}$ ha$^{-1}$) | *WUE* (kg mm$^{-1}$ha$^{-1}$) |
| Year (Y) | <0.0001 | <0.0001 | <0.0001 |
| Cultivar (C) | <0.0001 | <0.0001 | ns |
| sowing date (SD) | <0.0001 | <0.0001 | 0.001 |
| Plant density (PD) | 0.043 | 0.048 | <0.0001 |
| Y × C | <0.0001 | <0.0001 | ns |
| Y × SD | ns | ns | ns |
| Y × PD | ns | ns | 0.007 |
| C × SD | <0.0001 | <0.0001 | <0.0001 |
| C × PD | <0.0001 | 0.006 | 0.027 |
| SD × PD | ns | ns | ns |
| Y × C × SD | 0.017 | 0.017 | <0.0001 |
| Y × C × PD | ns | ns | ns |
| Y × SD × PD | ns | ns | 0.002 |
| C × SD × PD | ns | ns | ns |
| Y × C × SD × PD | ns | ns | ns |

Note: ns indicates no significant difference was observed.

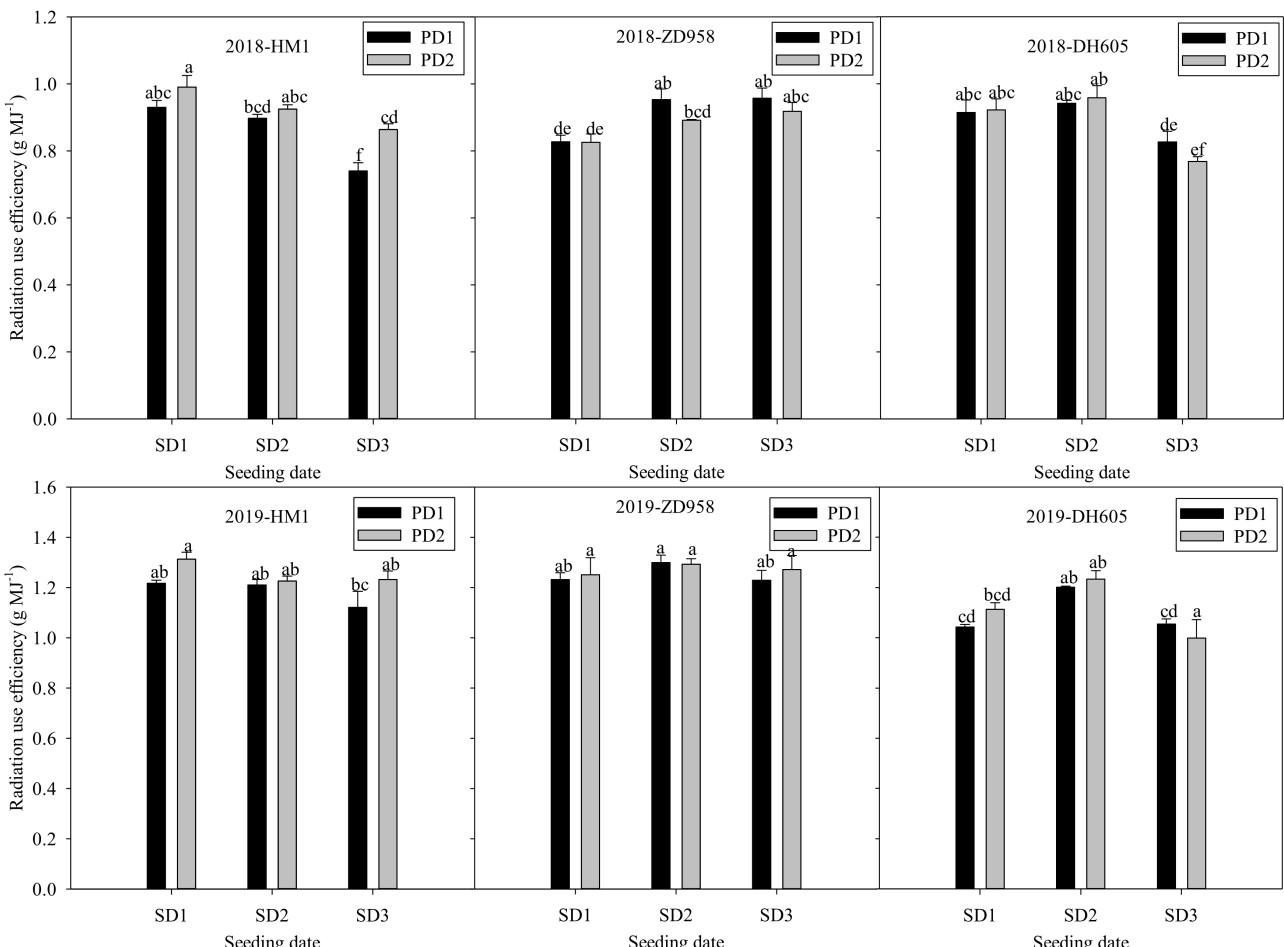

**Figure 3.** Effects of cultivar, sowing date and plant density on radiation use efficiency of summer maize. SD1, SD2, and SD3 represent the sowing dates of June 10, June 17, and June 24, respectively. PD1 and PD2 represent the plant densities of $6.75 \times 10^4$ plants ha$^{-1}$ and $8.25 \times 10^4$ plants ha$^{-1}$, respectively. Different lowercase letters above the columns within one year indicate a significant difference at $p < 0.05$.

### 3.4. Temperature Use Efficiency

Similar to the case of *RUE*, the effects of cultivar, sowing date, and plant density on *TUE* were significant, but their interactive effect on *TUE* was not obvious (Table 5 and Figure 4). ZD958 and HM1 had advantages over DH605 in *TUE*, and early sowing with higher plant density was beneficial to improve *TUE*. In both growing seasons, the highest *TUE* was observed for ZD958 at SD2-PD2; however, the optimal *TUE*s of the tested maize cultivars were not significantly different from each other.

### 3.5. Water Use Efficiency

Table 5 shows that the *WUE* of maize was significantly affected by sowing date and plant density, but the interactive effects of SD × PD and C × SD × PD on WUE were not obvious. The *WUE* of plants grown at SD2 was significantly higher than that of SD1 and SD3, and higher plant density was beneficial to improve *WUE* (Figure 5). In the 2018 growing season, the optimal *WUE* of all tested cultivars was observed at SD2-PD2, but the optimal *WUE* of DH605 was significantly higher than that of HM1 and ZD958. In the 2019 growing season, the optimal *WUE*s of HM1, ZD958 and DH605 were observed at SD3-PD2, SD2-PD1, and SD1-PD2, respectively, and the optimal *WUE* of HM1 was significantly higher than that of ZD958 and DH605.

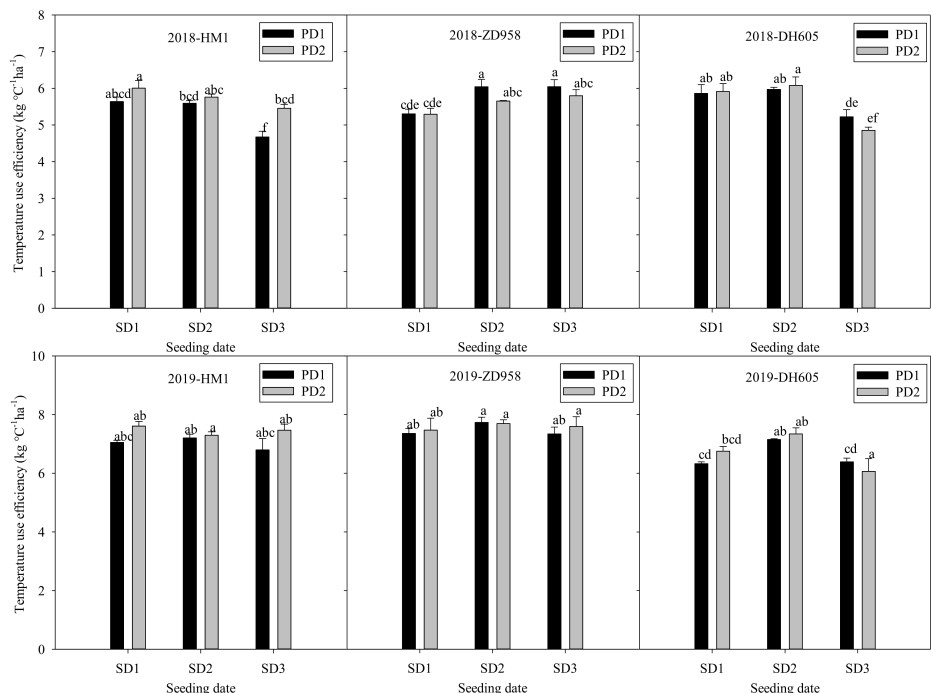

**Figure 4.** Effects of cultivar, sowing date and plant density on temperature use efficiency of summer maize. SD1, SD2, and SD3 represent the sowing dates of June 10, June 17, and June 24, respectively. PD1 and PD2 represent the plant densities of $6.75 \times 10^4$ plants ha$^{-1}$ and $8.25 \times 10^4$ plants ha$^{-1}$, respectively. Different lowercase letters above the columns within one year indicate significant differences at $p < 0.05$.

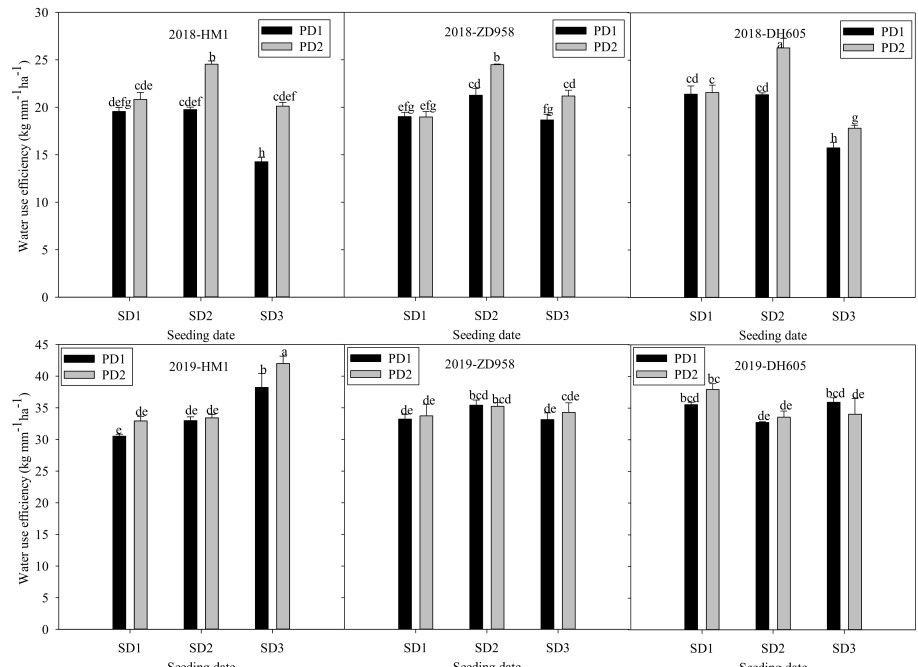

**Figure 5.** Effects of cultivar, sowing date and plant density on water use efficiency of summer maize. SD1, SD2, and SD3 represent the sowing dates of June 10, June 17, and June 24, respectively. PD1 and PD2 represent the plant densities of $6.75 \times 10^4$ plants ha$^{-1}$ and $8.25 \times 10^4$ plants ha$^{-1}$, respectively. Different lowercase letters above the columns within one year indicate significant differences at $p < 0.05$.

*3.6. Grain Moisture Content at Harvest*

Figure 6 shows that the grain moisture content of maize was significantly affected by the cultivar and seeding date. In both growing seasons, the grain moisture content of early-maturing cultivar HM1 at harvest was significantly lower than that of ZD958 and DH605, and delaying the sowing date significantly increased the grain moisture content. The lowest grain moisture contents of HM1, ZD958, and DH605 were 19.9%, 27.3%, and 25.2%, respectively.

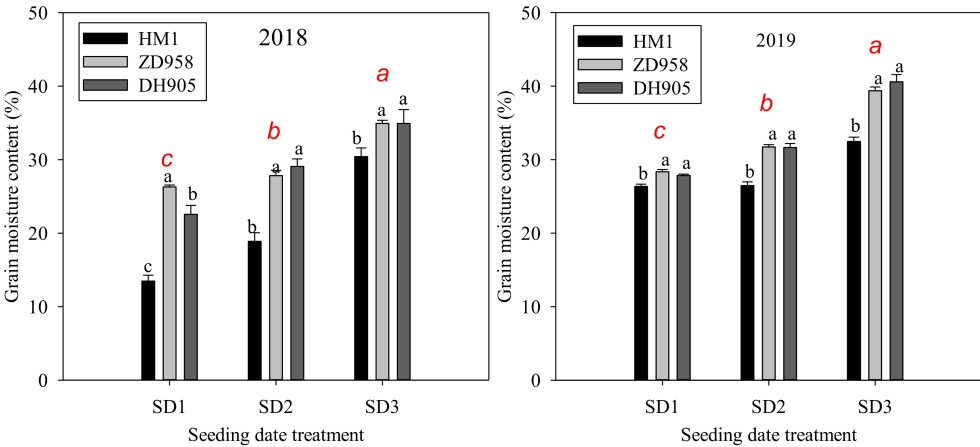

**Figure 6.** Grain moisture contents of tested cultivars at harvest under different seeding date. SD1, SD2, and SD3 represent the sowing dates of 10 June, 17 June, and 24 June, respectively. Different lowercase letters above the columns within the same sowing date indicate significant difference between cultivars. Different lowercase letters in red italics above the columns indicate significant difference between sowing dates.

## 4. Discussion

The rising temperatures caused by global climate change pose adverse effects for summer maize production in the Northern HHH Plain [34]. Prior studies have reported that optimal genotype × environment × management could be adopted as a strategy to increase grain maize productivity in the context of climate change [35]. Some have reported that adapted later-maturing cultivars can be effective at offsetting the negative impacts of climate warming on crop yield [36–39]. However, the present study showed that there was no significant difference in grain yield among cultivars with different maturity, suggesting that maize cultivars with contrasting maturity were not the main factor affecting the grain yield in the Northern HHH Plain in the context of climate change. Additionally, this study found that the effect of plant density on grain yield was not significant (Tables 2 and 4), indicating that plant density was also not a limiting factor on the grain yield of summer maize. Although reasonably increasing the plant density is an important agronomic practice for grain yield improvements [19,20], the present study showed that the effect of plant density on grain yield was not significant. The discrepancy may be associated with the density tolerance of tested maize cultivars in these studies, because selecting higher density-tolerant maize cultivars is the key to realize more grain yield under higher plant density [40]. Furthermore, the present results found that the effect of sowing date on grain yield was greater than cultivar and plant density effects, which is different from those of previous reports [10,25,41,42]. The main reason for the deviation may be related to the ecological environment or the cropping system in the study area. For example, some field experiments were conducted in a rain-fed cropping system, and some were conducted in a irrigated cropping system. Moreover, the present results showed that a delay in the sowing date decreased the grain yield in most cases, similar to the results of previous studies [11,12,16,43]. The North HHH Plain is characterized by limited solar-thermal resources [44]; therefore, the early sowing of summer maize is beneficial to the utilization of solar-thermal resources, and further improving yield. In this study, delaying the sowing

date significantly reduced the post-anthesis DMA, in line with prior studies [11] which reported that variations in grain yield resulting from different sowing dates were closely related to the DMA during the post-silking period. The present results revealed that the decreased grain yield of late sowing was associated with the lower DMA. In addition, delaying the sowing date significantly reduced the kernel weight, similar to previous results [12]. The lower kernel weight from seeds with a late sowing date was related to the limited photosynthetic source capacity [11]. The interactive analysis of cultivar, sowing date, and plant density indicated that the optimum grain yields of all tested cultivar were observed at SD1-PD2, and the difference in optimum grain yield among the tested cultivars with contrasting maturity stages was not significant. The present result suggest that plants grown at early sowing dates with higher plant density (75,000–82,500 plants ha$^{-1}$) exhibit improved maize grain yield in the North HHH Plain, which is consistent with the results from a previous study [45].

Improvements in the resource use efficiency of maize are often realized by optimizing cultivar selection and culture management practices [26,46]. The present results showed that the effects of cultivar, sowing date, and plant density on resource use efficiency were significant in most cases (Table 5). The *RUE* and *TUE* in HM1 and ZD958 were higher than that of DH605, suggesting that selecting early-maturing and medium-maturing cultivars was beneficial to *RUE* and *TUE*. However, the differences in WUE among the tested maize cultivars were not obvious, which is in contrast with the prior results [47], which reported that selecting a late-maturing maize cultivar could increase *WUE*. Manipulating the sowing date is one of the main management practices for improving crop yield and resource use efficiency [12–14,48]. Compared with a late sowing date, plants grown at early sowing dates and medium sowing dates had higher *RUE* and *TUE* in most cases, although the difference between early sowing date and medium sowing date was not significant, indicating that earlier sowing is beneficial for the *RUE* and *TUE* of summer maize. Generally, earlier sowing could promote the development and canopy closure of maize, and a rapid canopy closure was beneficial to *RUE* [47]. Reasonably increasing the plant density has been proven to be an effective agronomic practice for improving the resource use efficiency of maize [20,25]. In the present study, the effects of plant density on *RUE*, *TUE*, and *WUE* were significant (Table 5), and the *RUE*, *TUE*, and *WUE* at a plant density of 82,500 pl ha$^{-1}$ were higher than that of 67,500 pl ha$^{-1}$, which was similar to previous findings [26], revealing that reasonably increasing the plant density could realize the optimal *RUE* and *WUE* of maize. Usually, higher plant density not only promotes rapid canopy closure [47], but also increases the potential capacity of the crop canopy to capture resources [24]. In both growing seasons, the optimum *RUE* and *TUE* for HM1, ZD958, and DH605 were observed at SD1-PD2, SD2-PD2, and SD2-PD2, respectively, and the differences in optimal *RUE* and *TUE* among cultivars were not significant, suggesting that earlier sowing (i.e., maize planted before June 17) with a higher plant density could increase the *RUE* and *TUE*. However, the optimal *WUE* for the tested cultivars varied across years, which is probably associated with the different rainfalls during the two growing seasons (the precipitation during 2018 and 2019 growing seasons were 234.8 and 262.4 mm, respectively).

According to analysis of the single effects and interactive effects of cultivar, sowing date, and plant density on grain yield and resource use efficiency, the present results suggested that early sowing with reasonably dense planting benefits the grain yield, *RUE*, and *TUE* of summer maize in the Northern HHH Plain in view of mechanized maize grain harvesting becoming popular in the HHH Plain in recent years [49,50]. Suitable grain moisture contents is key to the mechanized grain harvesting of maize varieties, and previous study has confirmed that the grain moisture content best suited to mechanical maize grain harvesting ranges from 16.15% to 24.78% [51]. Therefore, developing maize cultivars characterized by faster grain dehydration rates are of priority in cultivar selection. Generally, the grain dehydration rate of early-maturing maize cultivars is faster than medium- or late-maturing cultivar at late growth period [50] The present results also showed that the early-maturing cultivar HM1 has the lowest grain moisture content (i.e.,

19.9%) at harvest (Figure 5). In brief, the early-maturing cultivar grown at the earlier sowing with reasonably higher plant density would be the optimal planting pattern for the Northern HHH Plain of China in future.

## 5. Conclusions

This study found that the effects of cultivar (with contrasting maturity) and plant density on grain yield of summer maize was not significant. Sowing date is the major factor affecting the grain yield in the Northern HHH Plain, and delayed sowing significantly decreased the grain yield of summer maize. However, RUE and TUE were significantly affected by cultivar, sowing date, and plant density; selecting early- and medium-maturing cultivars is beneficial to the improvement of RUE and TUE, and plants grown with early sowing with higher plant density increased the RUE and TUE. The interactive analysis of cultivar × sowing date × plant density analysis suggested that plants grown at early sowing with reasonable dense planting benefits grain yield and resource use efficiency. In order to adapt to mechanized maize grain harvesting, early-maturing cultivar with lower grain moisture at harvest would be the better choice. Therefore, adopting early-maturing cultivar grown at the earlier sowing with reasonably higher plant density would be the optimal planting pattern for the Northern HHH Plain of China in future.

**Author Contributions:** Conceptualization, L.Z. (Lichao Zhai); Data curation, L.Z. (Lihua Zhang), H.Y., M.Z., B.M., X.J. and J.J.; Formal analysis, L.Z. (Lichao Zhai), L.Z. (Lihua Zhang), B.M., R.X. and X.J.; Funding acquisition, R.X.; Investigation, L.Z. (Lichao Zhai), L.Z. (Lihua Zhang), H.Y., M.Z., J.Z. and J.J.; Methodology, B.M. and X.J.; Resources, H.Y. and J.Z.; Software, H.Y.; Validation, L.Z. (Lihua Zhang), B.M. and X.J.; Writing—Original draft preparation, L.Z. (Lichao Zhai); Writing—Review and editing, L.Z. (Lichao Zhai). All authors have read and agreed to the published version of the manuscript.

**Funding:** This study was supported by the Key Research and Development Program of Hebei Province (20326401D); National Key Research and Development Program of China (2016YFD0300106); Talents Project of Science and Technology Innovation, Hebei Academy of Agriculture and Forestry Sciences (C19R02-1-1).

**Institutional Review Board Statement:** Not applicable.

**Informed Consent Statement:** Not applicable.

**Data Availability Statement:** The data presented in this study are available on request from the corresponding author.

**Acknowledgments:** The authors thank the anonymous reviewers for their valuable comments and suggestions. The first author also thanks all the other co-authors' support and assistance all the time.

**Conflicts of Interest:** The authors declare no conflict of interest.

## Abbreviations

HHH: Huang–Huai–Hai; DMA, dry matter accumulation; WUE, water use efficiency; TUE, temperature use efficiency; RUE, radiation use efficiency; SD, sowing date; PD, plant density; HI, harvest index.

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
