# Peer review of "The Optimal Cultivar × Sowing Date × Plant Density for Grain Yield and Resource Use Efficiency of Summer Maize in the Northern Huang–Huai–Hai Plain of China"

_agriculture, doi:10.3390/agriculture12010007_

Round 1
Reviewer 1 Report
Review on: The optimal cultivar × sowing date × plant density for grain 2 yield and resource use efficiency of summer maize in the 3 Northern Huang-Huai-Hai Plain of China, by Lichao Zhai1, Lihua Zhang1,*, Haipo Yao1, Mengjing Zheng1, Jingting Zhang1, Bo Ming2, Ruizhi 5 Xie2, Xiuling Jia1,* Junjie Ji
The study examines the effect of two sowing dates and two plant densities over tree cultivar types of maize. Authors found that sowing date was the main factor affecting crop yield. Late sowing date had negative effects on maize productivity. Moreover, early sowing plus higher densities appear to enhance resource use efficiency; thus, authors suggest early sowing and high plant densities as the optimum planting pattern for the Huang-Huai-Hai Plain of China in present and future.
The manuscript is in general well written and have sound science and is relevant to be published in this journal. I think the manuscript needs minor revisions to be published. The manuscript needs at least two testable hypotheses in agreement with objectives. Some information is lost in introduction to follow in a better way the objectives, and particularly discussion and conclusions; for instance, climate change scenarios for the HHH Plain, i.e., how precipitation and temperature is expected to change at he middle and the end of this century, because I do not understand why the authors conclude early sowing and high plant densities are the best planting pattern for the future, when they are not imposing (or manipulating) future climate conditions in their experimental design, and they also do not explain how this planting pattern will improve corn yield in future climatic conditions, with exception of a more mechanized agriculture in the HHH Plain, but not climatic conditions. Explain based on data why did you conclude this.
The section of results is some hard to follow; I would prefer all data tables in figures and give available these tables as supplementary material. It is valuable to have numbers to compare with other studies, or as reference for future studies, but is easier to readers to see this information as figures.
ANOVA tables for all response variables should be as the primer information in results for checking significant effects and interactions of factors. Currently these tables are some lost along the manuscript.
Be more cautious about significance of factor effects when interactions are significant. In this study, many single effects depend on interaction of the three factors. Adjust discussion accordingly too.
Minor comments:
Include in results or material and methods the accumulation of PPT during the two seasons. Also, the accumulation of temperature (or heat units), radiation, etc., mainly for the three sowing dates. It should be known how different where climatic conditions between years, and among sowing dates. Then discuss about differences in these conditions; i.e. how much extra energy received the early sowing date plants. Should be this the cause of lower yield of late sowing date plants? In a climate change scenario of higher temperature, some crops grown in sub-optimal temperature requirements and sown at late dates could yield similar amounts of grain than crops sown at normal dates (https://doi.org/10.1016/j.fcr.2020.107903). How do you expect maize crops in the HHH Plain will respond to forecasted climate?
If the time series of soil moisture are available, please include this information in the firs figure.
How differences in PPT pattern between the two years could affect productivity? Specially in dry ecosystems, the PPT pattern affects water infiltration. Large PPT events could allow more water storage, and then more time without water stress for plants (see for instance https://doi.org/10.1016/j.gloplacha.2011.12.001; and https://doi.org/10.1016/j.agwat.2019.105728).
I expect authors enrich discussion with some of above data.
Other comments and suggestions are included into the manuscript.

Author Response
Responses to Reviewer#1’s comment
Comment 1: The manuscript is in general well written and have sound science and is relevant to be published in this journal. I think the manuscript needs minor revisions to be published. The manuscript needs at least two testable hypotheses in agreement with objectives. Some information is lost in introduction to follow in a better way the objectives, and particularly discussion and conclusions; for instance, climate change scenarios for the HHH Plain, i.e., how precipitation and temperature is expected to change at he middle and the end of this century, because I do not understand why the authors conclude early sowing and high plant densities are the best planting pattern for the future, when they are not imposing (or manipulating) future climate conditions in their experimental design, and they also do not explain how this planting pattern will improve corn yield in future climatic conditions, with exception of a more mechanized agriculture in the HHH Plain, but not climatic conditions. Explain based on data why did you conclude this.
Answer: Thank you for your comment. First, we have added a testable hypothesis in the introduction section which in agreement with the objectives.
The Northern HHH Plain is characteristic with a area of limited light-thermal resource, frequent extreme weather events affect local maize production. The main objectives of this study were to (1) investigate the individual and combined effects of cultivar, sowing date, and plant density on grain yield and resource use efficiency of summer maize, and (2) determine the optimal cultivar × sowing date × plant density for grain yield and resource use efficiency of summer maize in the Northern HHH Plain, and further provide provide technical support for local maize production for the present time, but not to provide technical support for the maize production at the middle and the end of this century. Therefore, the present study
I think the future climate condition is hard to predict except the warming climate, the present study mainly focus on problems of the current regional summer maize production (i.e.lower plant density,low light-thernal resource use efficiency, et al.)
We concluded that early sowing combined with higher plant densities are the best planting pattern for grain yield in the HHH plain, because this planting pattern increased the harvest ear number, and dry matter accumulation, we think this is the main reason for the improved grain yield, and we have discussed it in the discussion section.
Comment 2: The section of results is some hard to follow; I would prefer all data tables in figures and give available these tables as supplementary material. It is valuable to have numbers to compare with other studies, or as reference for future studies, but is easier to readers to see this information as figures.
Answer:Thank you for your comment. However, we don’t quite agree with your comment that convert the all table data into Figures, if so , another seven combined figures (i.e. grain yield, ear number; kernels per ear, kernel weight, pre-anthesis DMA, post-anthesis DMA, and total DMA will added in this manuscript, and there will be 3 or 4 figures for the result section of “grain yield and yield components” and “dry matter accumulation, more figures will make these result section not be concentrated.
Comment 3: ANOVA tables for all response variables should be as the primer information in results for checking significant effects and interactions of factors. Currently these tables are some lost along the manuscript.
Answer: Thank you for your comment and we agree with this comment. In the revised manuscript, we have replaced the ANOVA tables in the result section, and made it as the primer information in results, which will benefits readers to check significant effects and interactions of factors.
Comment 4:Be more cautious about significance of factor effects when interactions are significant. In this study, many single effects depend on interaction of the three factors. Adjust discussion accordingly too.
Answer: Thank you for your comment. However, we don’t quite agree with your comment that many single effects depend on interaction of the three factors. Usually, if the interactive effect of all factors was significant, the effect of single factors were all significant. However, if all the single factors were significant, its interactive effect may not necessarily significant.
Minor comments:
Comment 5: Include in results or material and methods the accumulation of PPT during the two seasons. Also, the accumulation of temperature (or heat units), radiation, etc., mainly for the three sowing dates. It should be known how different where climatic conditions between years, and among sowing dates.
Then discuss about differences in these conditions; i.e. how much extra energy received the early sowing date plants. Should be this the cause of lower yield of late sowing date plants? In a climate change scenario of higher temperature, some crops grown in sub-optimal temperature requirements and sown at late dates could yield similar amounts of grain than crops sown at normal dates (https://doi.org/10.1016/j.fcr.2020.107903). How do you expect maize crops in the HHH Plain will respond to forecasted climate?
Answer: the accumulation of PPT during 2018 and 2019 growing season were 244.8 mm and 262.4 mm, respectively, and we have added it in the material and method section in the revised manuscript. In addition, the accumulation of temperature of SD1, SD2, SD3 in 2018 were about 2822.8, 2763.6, 2686.3℃, respectively; and the accumulation of temperature of SD1, SD2, and SD3 in 2019 were about 2763.5, 2696.7, 2613.5℃, respectively. Although, the accumulation of early sowing date plants were higher than that of other sowing date( the extra temperature accumulation was varied from 38.8 to 150.0 ℃). We agree with your comment that the the lower temperature accumulation is one main reason for its lower grain yield. In a climate change scenario of higher temperature, we think summer maize plants sown in late times could yield similar grain yield just as the plants sown at normal date in the Northern HHH plain, because the higher temperature will reduce the growing days to reach the accumulative temperature required for some maize cultivars.
Comment 6: If the time series of soil moisture are available, please include this information in the firs figure.
Answer: Thank you for your suggestion. However, the time series of soil moisture are not available in the present study, it could be considered in our further studies.
Comment 7: How differences in PPT pattern between the two years could affect productivity? Specially in dry ecosystems, the PPT pattern affects water infiltration. Large PPT events could allow more water storage, and then more time without water stress for plants (see for instance https://doi.org/10.1016/j.gloplacha.2011.12.001; and https://doi.org/10.1016/j.agwat.2019.105728).
Answer: Thank you for your concern on our manuscript. the PPT of 2018 and 2019 growing seasons were 234.8 mm and 262.4 mm, respectively; the PPT during the two growing seasons are normal rainfall years. Usually, two irrigation operation are required during the summer maize growing season in the Northern HHH Plain. In our experiment, two irrigation were also applied during each growing season according to the rainfall conditions. We agree with your comment that the PPT pattern affects water infiltration, and large PPT events could allow more water storage, which could effectively avoid the water-stress for plants. However, the Northern HHH Plain is characterized with continental monsoon climate, and is not a rainfed area, irrigation was usually conducted by farmer according the rainfall conditions, so maize will not suffer drought stress due to the low-level rainfall, just as some rainfed area.
Other comments and suggestions are included into the manuscript.
1.Comment: Line 172 How frequent was it?
Answer: it was measured with three replicates, we have mentioned it in the revised manuscript.
- Comment:Line 176,I think this reference does not support this statement; maybe 33 do.Did large PPT events produce runoff in the site?
Answer: We don’t quite agree with you on this comment, the research of this reference cited here was also conducted in the Northern HHH Plain. During the 2018 and 2019 growing season, the PPT was 234.8 and 262.4 mm, it did not produce the surface runoff in the experimental site.
- Comment:Line198, I would like to see the ANOVA table .
Answer: Table 2 is the ANOVA table in the revised manuscript.
- Comment: Line 200, but, it was not significant?
Answer: Yes. Although the grain yield under PD2 was higher than that of PD1, the difference between PD1 and PD2 was not significant.
- Comment:Line 203-205,Was the interaction significant?
Answer: The interactive analysis of cultivar, sowing date, and plant density showed that only significant difference was observed for KW, and it was included in this result section.
- Comment:Table 4, give meanings of these initials in table description.
Answer: Thank you for your suggestion, we have given the meanings of these initials in Table description in the revised manuscript.
- Comment:Line 237, what it means?
Answer: It means that the single effect of cultivar, sowing date, and plant density on RUE was significant, but the interactive effect of cultivar, sowing date, and plant density was not significant.
- Comment:Line 264-265, this should be in discussion section
Answer: We agree with this comment, and have deleted this sentence in this paragraph, and discussed it in the discussion section.
- Comment: Line267-269, this should be in discussion section
Answer: We agree with this comment, and have deleted this sentence in this paragraph, and discussed it in the discussion section.
- Comment: Line292-293, However, there is an interaction of PD with cultivar.
Answer: Indeed, the interactive effect of plant density and cultivar on grain yield was significant. However, this sentence here mainly describes the single effect of plant density on grain yield.
- Comment:Line 296-297, So, these results are in agreement with the number of kernels, or the weight of 300 grains, or the number of ears? I guess any of these should be lower for higher than lower densities.
Answer: Usually, in field maize density experiments, the harvested ear number per unit area of higher plant density was higher than that of lower plant density, by contrast, the kernel number per ear and kernel weight and kernel weight of higher plant density was lower than that lower plant densities.
- Comment: Line 303, Which ecological variables or cropping systems?
Answer: our results was different from previous studies, because these studies were conducted under different cropping systems. For example, some were conducted under rain-fed cropping system, and our study was conducted under irrigated cropping system.
- Comment: Line 347-348, You should indicate how much PPT occurred each year
Answer: We agree with your comment, and have made some specific change in the revised manuscript.
- Comment: Line359-361, Check this sentence, I think something is lost
Answer: In order to make this sentence more clear for readers, we have made a revision on this sentence in the revised manuscript.
- Comment: Line 368, why?
Answer: First, the northern HHH Plain is characterized with winter wheat- summer maize double cropping system, and summer maize was usually grown from the middle June to late September, the growing period of summer maize was relative narrow and it is only about 100 days, so only select early maturing cultivar planted at early sowing date can realize the mechanized grain harvesting and it grain yield. Moreover, as it is known to all, the grain yield potential of early maturing cultivar was lower than the medium or late maturing cultivar, in order to improve the grain yield, increasing plant density reasonably is an key to improve the grain yield, because it has been confirmed by previous studies.

Reviewer 2 Report
This study aims to assess the individual and combined effects of cultivar, sowing date, and plant density on grain yield and resource use efficiency of summer maize in Huang–Huai–Hai Plain. The structure and and conclusions are consistent with aiming of the MS.
However, I have some reservations on how the data was collected and how the results were discussed.
Sections numbering needs to be revised.
Abstract: The text is somewhat confusing. The authors should make it clearer, highlighting the main outcomes of the study in a more concise form.
Introduction: This section can be improved; Past studies were conducted this subject. The main findings of those studies need to be outlined in this section.
2.1. Experimental site: A map with the location of the study area would be appreciated. Also, data on the soil physical properties are required (field capacity, wilting point, ....)
2.2. Experimental design and field management: Figure 2 isn't cited on the text.
Nonetheless, the publication of the MS should be reconsidered after minor revisions.
Author Response
Responses to Reviewer2#’s comments
- Comment: Sections numbering needs to be revised.
Answer: We agree with this comment, and have made the specific change in the revised manuscript.
- Comment: Abstract: The text is somewhat confusing. The authors should make it clearer, highlighting the main outcomes of the study in a more concise form.
Answer: Thank you for your suggestions. The main objectives of our study were to (1) investigate the individual and combined effects of cultivar, sowing date, and plant density on grain yield and resource use efficiency of summer maize, and (2) determine the optimal cultivar × sowing date × plant density for grain yield and resource use efficiency of summer maize in the Northern HHH Plain. So, the single effect of cultivar, sowing date, and plant density, and the interactive effects of cultivar × sowing date × plant density on grain yield and resource use efficiency is necessary to make it clear for readers in the abstract section. We think it might a little difficult to make it more concise, or some key information will be lost.
- Comment: Introduction: This section can be improved; Past studies were conducted this subject. The main findings of those studies need to be outlined in this section.
Answer: Thank you for your comments. However, previous studies related to the effect of cultivar, sowing date, and plant density have been cited in the introduction section in our manuscript, including those published in Field Crops Research, Agricultural Water Management, Journal of Integrative Agriculture in recent years, please refer to the reference section.
- Comment: Experimental site: A map with the location of the study area would be appreciated. Also, data on the soil physical properties are required (field capacity, wilting point, ....)
Answer: Thank you for your suggestion. According to your suggestion, a map with the location of the study area was added in the revised manuscript, please see Figure 1. the top soil bulk density and soil porosity was added in the material and method section in the revised manuscript.
- Comment:Experimental design and field management: Figure 2 isn't cited on the text.
Answer: Actually, Figure.2 was cited in the result section

This manuscript is a resubmission of an earlier submission. The following is a list of the peer review reports and author responses from that submission.